# Aerogel Perfusion-Prepared h-BN/CNF Composite Film with Multiple Thermally Conductive Pathways and High Thermal Conductivity

**DOI:** 10.3390/nano9071051

**Published:** 2019-07-23

**Authors:** Xiu Wang, Zhihuai Yu, Liang Jiao, Huiyang Bian, Weisheng Yang, Weibing Wu, Huining Xiao, Hongqi Dai

**Affiliations:** 1Jiangsu Co-Innovation Center of Efficient Processing and Utilization of Forest Resources, Nanjing Forestry University, Nanjing 210037, China; 2Department of Chemical Engineering, University of New Brunswick, Fredericton, NB E3B 5A3, Canada

**Keywords:** CNF aerogel, perfusion, h-BN, thermal conductivity, electrical insulation

## Abstract

Hexagonal boron nitride (h-BN)-based heat-spreading materials have drawn considerable attention in electronic diaphragm and packaging fields because of their high thermal conductivity and desired electrical insulation properties. However, the traditional approach to fabricate thermally conductive composites usually suffers from low thermal conductivity, and cannot meet the requirement of thermal management. In this work, novel h-BN/cellulose-nano fiber (CNF) composite films with excellent thermal conductivity in through plane and electrical insulation properties are fabricated via an innovative process, i.e., the perfusion of h-BN into porous three dimensional (3D) CNF aerogel skeleton to form the h-BN thermally conductive pathways by filling the CNF aerogel voids. When at an h-BN loading of 9.51 vol %, the thermal conductivity of h-BN/CNF aerogel perfusion composite film is 1.488 W·m^−1^·K^−1^ at through plane, an increase by 260.3%. The volume resistivity is 3.83 × 10^14^ Ω·cm, superior to that of synthetic polymer materials (about 10^9^~10^13^ Ω·cm). Therefore, the resulting h-BN/CNF film is very promising to replace the traditional synthetic polymer materials for a broad spectrum of applications, including the field of electronics.

## 1. Introduction

The rapid development of miniaturization, integration, and high-power electronic devices brings with it higher requirements for their effective heat-dissipation properties [1,2,3]. Considerable heat will be accumulated and make an electronic device heat up during the working process if the heat cannot be transferred out rapidly. This will affect the operation quality considerably, and compromise the lifetime and reliability of electronic devices significantly [2,4,5], or even cause an explosion and other serious accidents [6,7]. It is vital for the electronic and electrical industry to prevent heat accumulation in the electronic devices by exporting the heat in a timely way [6,8,9,10]. Synthetic polymers have been widely used in encapsulation of electronic devices as traditional insulating packaging materials [11,12,13]. However, the thermal conductivities of synthetic polymer materials are usually very low (about 0.1~0.5 W·m^−1^K^−1^ at room temperature) [7,14,15,16,17], and polymer itself also has aging issues leading to embrittlement, such as polyimide (PI) [18], polyimide 11 (PI 11) [19], and polyethylene (PE) [20] films. Hence, it is an inevitable trend for the electronic and electrical equipment industry to replace synthetic polymer materials with green and eco-friendly insulating materials with high thermal conductivity. Thermal insulating paper used in electronic devices is usually made from plant fibers [21], which have excellent thermal stability and electrical insulation properties but poor thermal conductivity [22]. Metal oxides (aluminum oxide, zinc oxide, beryllium oxide, etc.), ceramic and metal nitride (boron nitride [23,24], aluminum nitride, etc.) have good thermal conductivities and electrical insulation properties [25,26], but cannot be folded and crimped [27]. Obviously, insulating composites with excellent thermal conductivity and flexible folding properties can be prepared by the combination of cellulose fibers and thermally conductive inorganic metallic compounds [23]. Boron nitride (BN) as a highly thermally conductive and electrical insulating two dimensional (2D) material has attracted great interest in recent years due to its large and direct band gap, relatively low price, resistance to oxidation, and high elastic modulus [28,29,30], which is one of the most promising 2D thermal conductive fillers [4]. Micro-nano hexagonal boron nitride (h-BN) exfoliated from commercial h-BN, with the thinner thickness and larger specific surface area, has a higher thermal conductivity which is widely used in composites [4,31]. However, the thicknesses of electronic insulating film materials such as polyimide films are often below 125 μm [32], and thinner film is always desirable. Apparently, it is rather difficult for fiber-based film to reach a similar thickness to that of synthetic polymeric ones as plant fiber itself has average diameter about 15~30 μm [33]. Moreover, it is hard to form thermally conductive pathways between the two sides of the composite because of the barrier between fibers and inorganic metallic compound particles [34,35]. To address this problem, perfusing h-BN into cellulose-nano fiber (CNF) (100~2000 nm in length, <100 nm in diameter) aerogel originating from plant fibers was proposed in an attempt to create multiple thermally conductive pathways of the composite film [36]. This method not only overcomes the drawback of high resistance between ordinary plant fibers and thermal conducting particles, but also makes the film thinner, leading to a flexible, foldable, high temperature and aging-resistant insulating material.

In our previous work [6], the BN/CNF composite film was fabricated by mixing CNF and BN to fabricate BN/CNF aerogel, and then filled with CNF dispersion. The drawback is that the existence of CNF blocked the full contact between BN. In this study, we first report a novel strategy for the fabrication of h-BN thermally conductive pathways on a 3D cellulose aerogel skeleton. The micro-nano h-BN with excellent thermal conductivity and electronic insulation performance was selected as thermal conductivity material. The highly thermally conductive composite films were then prepared using a perfusion method, i.e., impregnating CNF aerogels into a certain amount of h-BN suspension with ultrasonic-assisted mixing so that the h-BN closely connected with each other and thermal conductive pathways can be formed in the network gap of CNF aerogel. The resulting green, eco-friendly and high thermally conductive and electrical-insulating material is of great potential to substitute synthetic polymer material for various electronic devices.

## 2. Experimental Section

### 2.1. Materials

The cellulose used in this work was bleached softwood pulp (Ilim Pulp Co. Ltd., Bratsk, Russia). The hexagonal boron nitride (h-BN) powders with an average diameter of 1~2 μm were obtained from Aladdin reagent Co. (Shanghai, China). 2,2,6,6-tetramethylpiperidine-1-oxyl (TEMPO) was purchased from Jiana Chemical Co., Ltd. (Changzhou, China). Isopropyl alcohol, sodium bromide were supplied from Nanjing Chemical reagent Co. (Nanjing, China); sodium hypochlorite solution, with an effective chlorine content of 12%, was purchased from Sinopharm group Co., Ltd. (Shanghai, China). A cationic papermaking-grade poly(aminoamide) epichlorohydrin resin (PAE) solution (Tianma Specialty Chemicals Co., Suzhou, China) was used without any purification. Polyvinylidene fluoride (PVDF) membrane with a pore size of 0.22 μm was obtained from Aladdin reagent Co. (Shanghai, China).

### 2.2. Preparation of Cellulose-Nano Fiber (CNF)

Bleached softwood pulp with mechanical pretreatment (5 g) was suspended in 450 mL of deionized water and 50 mL of NaClO. Then, TEMPO (0.08 g) and NaBr (0.8 g) were added into the solution and stirred at 250 rpm at 25 °C for 8 h. The pH was adjusted to about 10.0~10.5 by HCl (20% *v*/*v*) and NaOH (0.5 mol·L^−1^). Then, the reaction was terminated by adding ethyl alcohol (10 mL), followed by adjusting pH to 7 with HCl (1 mol·L^−1^). The suspension was dialyzed using a dialysis bag with MW-CO 12000~14000 (D) after centrifuging at 5000 rpm 3 times. Eventually, the suspension was treated with a homogenizer (FB-110X, ShangHai LiTu Mechanical Equipment Engineering Co. Ltd., Shanghai, China) under a pressure of 600 bar for 10 cycles.

### 2.3. Exfoliation of Hexagonal Boron Nitride (h-BN)

The liquid phase exfoliation of h-BN was performed according to the work reported previously [4], as shown in Appendix A. Briefly, a certain amount of h-BN powders was added into a mixture solvent of isopropanol and deionized water (1/1). The suspension was sonicated in a sonication bath (X0-650, Xianou Instruments Ltd., Nanjing, China) for 8 h with a frequency of 200 kHz and a 520 W output power. The resulting dispersion was centrifuged at speed of 3000 rpm for 20 min, the supernatants were dried at 80 °C for 24 h to obtain micro-nano sized h-BN power.

### 2.4. Preparation of CNF Aerogel

Cellulose aerogels were fabricated by a typical method. In brief, 33.4 g, 0.5 wt % of CNF and 0.83 g, 0.1 wt % of PAE dispersion were mixed in an ice bath under the ultrasonic condition at 300 W output power (X0-650, Xianou Instruments Ltd., Nanjing, China) [37]. To prevent flocculation of CNF, PAE should be added drop by drop. Then the mixture slurry was poured into polystyrene petri dishes (diameter 65 mm) and frozen in liquid nitrogen. The CNF aerogels were obtained after freeze-drying at −80 °C, 15 Pa for 72 h in a FD-1C-80 freeze dryer (Shanghai Yuming Instruments Ltd., Shanghai, China). CNF aerogels with enhanced wet strength were obtained after heated at 105 °C for 30 min to induce cross-linking.

### 2.5. h-BN/CNF Composite Films Preparation

In this study, h-BN/CNF composite films were prepared by the blending method and aerogel perfusion method, as shown in Figure 1.

#### 2.5.1. Preparation of h-BN/CNF Blended Composite Film

CNF was mixed with the h-BN suspension in the beaker. Various amounts of h-BN (see Table 1) were added with stirring for 30 min, and the concentration of the mixed suspension was adjusted to 0.5 wt %. Then, the mixture slurry was poured into polystyrene petri dishes (diameter 65 mm) and transferred into a blast oven at 25 °C for 7 h to obtain the gel. The mixture after preliminary dehydration was placed between two PVDF membranes to compact under 0.5 MPa pressure. Finally, the composite films were calendered (YYG-300, Dandong Yutai Instruments Ltd., China) under 1 MPa for 10 times. All the samples were conditioned at 25 °C and 50% relative humidity (RH) for 72 h before measurements.

#### 2.5.2. Preparation of h-BN/CNF Aerogel Perfusion Composite Film

The h-BN/CNF aerogel perfusion films were fabricated using ultrasonic-assisted impregnation of h-BN suspension. The h-BN loading was controlled through changing the h-BN solution volume. After ultrasonic infiltration for 15 min, 30 min, 45 min, 60 min, 75 min, and 90 min, the samples were transferred into a blast oven at 25 °C for 5 h to remove a part of the water. The aerogel composites after preliminary dehydration were placed between two PVDF membranes to compact under 0.5 MPa pressure. Finally, the composite films were calendered (YYG-300, Dandong Yutai Instruments Ltd., Dandong, China) under 1 MPa for 10 times. All the samples were conditioned at 25 °C and 50% RH for 72 h before measurements.

### 2.6. Characterization

The morphologies of h-BN and CNF were characterized using a dimension edge atomic force microscope (AFM) (Bruker, Germany) in tapping mode at 300 kHz. The morphologies of the h-BN, CNF aerogel, and h-BN/CNF composite films were observed using an environmental scanning electron microscope (SEM) (Quanta-200, FEI, Hillsboro, OR, USA). The thermal stability of the composite films were measured via thermogravimetric (TG) analysis using a thermogravimetric analyzer TGA (Q5000IR, TA instruments, New Castle, DE, USA). The volume resistance was tested with an insulation resistance tester (ZC36, Shanghai Jingke Industrial Co., Shanghai, China) at 25 °C and 50% humidity. Thermal conductivity was measured through the laser flash technique (LFA 467 HT HyperFlash, NanoFlash, Netzsch, Seelze, Germany). Every film was cut into a square with size of 10 × 10 mm, the thermal diffusion coefficient of the sample was measured at 25 °C under the N_2_ atmosphere of 50 mL/min. The specific heat of the sample was obtained by DSC 204 F1 Phoenix (Netzsch, Germany); the density was measured using a real density meter with exhaust method. Thermal conductivity *K* (W·m^−1^·K^−1^) was calculated as a multiplication of density (*ρ*, g·cm^−3^), specific heat (*Cp*, J·g^−1^·K^−1^), and thermal diffusivity (*α*, mm^2^·s^−1^); namely: *K*(T) = *α*(T) × *Cp*(T) × *ρ*(T). The composite film sample was placed in the oven of 80 °C with constant temperature for 6 h, and then transferred to the platform on the cold plate at room temperature (around 19 °C). The temperature of the composite films were recorded by an infrared thermal-graph (Ti45, Fluke, Madison, WI, USA).

## 3. Results and Discussion

### 3.1. Preparation of h-BN/CNF Composite Film

In this study, CNF was prepared by the combination of TEMPO/NaClO/NaBr oxidation and high-pressure homogenization, and the carboxylate content of CNFs was 1.587 mmol·g^−1^. Figure 2a shows the AFM image of CNF with length of 300~1000 nm and Figure 2b is the corresponding height distribution measured from AFM topography, which shows the diameters of CNFs around 5~7 nm. The diameter of CNFs was 5~7 nm (<100 nm) and exhibited a Tyndall effect in Appendix A [38]. This endows good flexible and folding properties in the composite film because of the high aspect ratio of CNFs (around 40~200). Appendix A present the images of the cellulose aerogel, shaped like a “white foam”. Hydrogen boding governs the strength of CNF aerogel, which tends to be broken when exposed to water, leading to poor aerogel strength. Therefore, a wet strength agent, i.e., PAE, was added to CNF solution to endow the CNF aerogel water resistance. Figure 2c,d present the SEM images under different magnifications of the CNF aerogel, demonstrating that a typical 3D porous structure with pore diameter of 5~20 μm was created.

In order to increase thermal transfer efficiency of h-BN, theoretically, the thermal conductive network should be constructed as much as possible. In this case, the thickness of h-BN sheets should be kept as thin as possible and the specific surface area of h-BN should be large. Exfoliation is a common method to obtain the micro-nano h-BN [39]. Therefore, it is desirable to increase the contact area and decrease the inter-layered gap between h-BN for the conductive network construction, leading to the relatively gentle phonon scattering at the interfaces of h-BN and thus the high thermal transfer efficiency [2]. Appendix A presents the schematic of exfoliation of h-BN. Few-layer h-BN was prepared by sonication-assisted liquid-phase exfoliation using a mixture of isopropanol and deionized water as solvent for both dispersion and exfoliation. Under the intense vibration of ultrasonic wave, the polar molecules of isopropanol permeate into the h-BN film layer, and large size h-BN particles were separated. After centrifuging and collection, a uniform and stable h-BN suspension was obtained, as shown in Figure 2h. When irradiated by red laser beam, an obvious pathway can be observed, resulting in the Tyndall phenomenon [40]. The morphologies of the h-BN were observed using SEM and AFM. The morphology of the commercial h-BN is shown in Appendix A. The original h-BN showed the scale-like shape, with lateral sizes in the range of 1~2 μm and the thickness of about dozens of layers. After exfoliation, the transverse dimensions and sheet thicknesses were calculated according to the AFM topographic images of the exfoliated h-BN as shown in Figure 2e,f. Figure 2g is the corresponding height distribution measured from AFM topographic data of h-BN. We can see that the length and height of h-BN were about 400~1000 nm and 50~130 nm, respectively. Compared to our previous work [6], the transverse dimensions and thickness of boron nitride nano-sheets (BNNS) were about 500 nm and 5nm (Appendix A). This indicates that with the increase of the centrifugal speed, the size of the h-BN in supernatants became smaller. All the data suggested that the h-BN was exfoliated successfully, which will increase the heat conductivity.

Figure 1 presents the synthesis pathways of h-BN/CNF composite films by simple blending and aerogel perfusion methods. In brief, the h-BN/CNF blended composite film was prepared by mixing CNF with h-BN in suspension; after drying and calendering, the h-BN/CNF blends were obtained. The h-BN/CNF aerogel perfusion films were fabricated using ultrasonic-assisted impregnation of h-BN suspension into CNF aerogel. After ultrasonic infiltration, h-BN was packed into the 3D network of CNF aerogel and formed thermally pathways. After drying and calendering, the h-BN/CNF aerogel perfusion film was fabricated.

Figure 3a shows an image of h-BN/CNF composite film using the perfusion method with the h-BN content of 23.08 wt %. Figure 3b presents the optical photograph of a “paper crane” folded from the composite film shown in Figure 3a. This demonstrated good folding and crimping performance of the aerogel perfusion-prepared composite film. The surface morphologies of the h-BN/CNF composite films are clearly exhibited in Figure 3c,d. Comparing two morphologies, it can be seen that the composite film prepared by perfusion method bears more h-BN particles than h-BN/CNF blended composite film. This is due to the fact that the blended method makes the mixture of CNF and h-BN more uniform, whereas the perfusion method would have more h-BN trapped on the surface of the CNF aerogel. The cross-section images of the h-BN/CNF composite film are shown in Figure 3e,f. As shown in Figure 3e, after impregnation by h-BN suspension, the pores of CNF aerogel were fully filled with h-BN. The layered structure of h-BN and CNF could also be observed, but the exact shape of the nanoscale h-BN could not be revealed due to the limited magnification. However, those in white pathways indeed represent the h-BN thermally conductive pathways, which are framed by blue rectangles in the Figure. These features would be beneficial for enhancing the thermal conductivity of the h-BN/CNF composite film material. According to the morphologies of CNF, CNF aerogel, h-BN and the cross-sectional SEM image of h-BN/CNF aerogel perfusion composite film, the proposed thermal defusion mechanism is illustrated in Figure 4, in which the white filaments represent CNFs, h-BN is shown in a yellow hexagon, and the thermally conductive pathways formed by h-BN contacting each other are represented with red lines. As can be seen from Figure 3e, the thermally conductive pathways are vertical, so the prospective thermal defusion direction is upright, as indicated by the red arrows. Meanwhile, no obviously thermally conductive pathways were observed in h-BN/CNF blended composite film (Figure 3f).

### 3.2. Thermal Stability of h-BN/CNF Aerogel Perfusion Composite Films

The decomposition temperature of h-BN/CNF aerogel perfusion-prepared composite film started from 200 °C while pure CNF film degradation began at 175 °C. The lower onset degradation temperature of CNFs is due to the existence of carboxyl groups on the surface of CNFs [41,42]. The TG and the corresponding derivative thermogravimetry (DTG) curves of the h-BN/CNF aerogel perfusion composite film are shown in Figure 5. For the pure CNF film, the initial decomposition temperature and the maximum decomposition rate were 175 °C and 272 °C, respectively. The initial decomposition temperature of pure CNF film was lower than that reported elsewhere [43], possibly because the voids in the film forming from aerogels reduce the thermal stability of the pure CNFs film. The initial decomposition temperature and the maximum decomposition rate of the film gradually increased with the increase of h-BN content in the composite film. When the h-BN contents were 16.67 wt % and 23.08 wt %, the initial decomposition temperature of the composite film increased from 175 °C to 200 °C and 218 °C, respectively; and the temperatures corresponding to the highest rate of decomposition temperature increased from 272 °C to 279 °C and 292 °C, respectively. It has been reported that pure h-BN exhibits high thermal stability on heating up to 800 °C under N_2_ atmosphere [44], indicating that h-BN can effectively slow down the decomposition rate of composite film. The h-BN added to the composite film enhances the heat resistance and barrier effect because of its high heat capacity and thermal conductivity; meanwhile, this prevents CNF components in composite film from being decomposed promptly. Moreover, with the layered structure of h-BN/CNF aerogel composite film, h-BN closely connected with each other and provided a tortuous path for the diffusion of gas molecules and significantly reduced the permeation rate of gas. Herein, the oriented h-BN platelets acted as a physical barrier and delayed the escape of degradation products, which finally promoted the thermal decomposition temperature to move towards high temperature [31]. Thus, the thermal stability of the composites was improved.

### 3.3. Thermal Conductivity and Electrical Properties of h-BN/CNF Composite Films

To investigate the thermal conductive properties of the h-BN/CNF composite films, the specific heat, density and thermal diffusivity of the blended composite film and aerogel perfusion composite film were tested respectively in the experiment, as shown in Appendix A. According to Appendix A, the specific heat of blended composite film was very close to the aerogel perfusion composite film when the h-BN loading was the same. This was probably due to the fact that the heat absorbed by the material remained the same with the temperature rising so long as the composition of materials is identical, which was hardly affected by the internal structure of the material [45]. However, the thermal diffusivity of aerogel perfusion composite film was higher than that of blended composite film with the same content of h-BN. The higher thermal diffusivity rate was probably attributed to more heat transfer passages formed in the aerogel perfusion composite film; or less heat transfer resistance between h-BN and CNF.

Taking the content of h-BN as abscissa and the thermal diffusivity as ordinate, the linear distribution line is shown in Figure 6a. As can be seen, the thermal diffusivity of the film was 0.268 mm^2^·s^−1^ without h-BN, regardless of preparation method. After loading with the same amount of h-BN, the thermal diffusivity of composite films prepared by the aerogel perfusion method was greater than that of composite films prepared by blending method. The correlation coefficient (R^2^) is an indicator to the degree of affinity; the larger the R^2^, the higher the correlation between independent and dependent variables. It is of interest that starting from 0 wt%, the thermal diffusivity of the h-BN/CNF composite film increased linearly with the h-BN content. Comparing two types of composite films at the same loading, we found that R^2^ of blended composite film was higher than that of aerogel perfusion composite film, implying that the thermal diffusivity of the blended film increases by a much similar degree for every increment of h-BN content.

Inserting the data in the table into the Equation (1) leads to the thermal conductivity of the composite films, which is defined as:*K*(T) = *α*(T) × *Cp*(T) × *ρ*(T)(1)
where *K* (W·m^−1^K^−1^), *ρ* (g·cm^−3^), *Cp* (J·g^−1^K^−1^) and *α* (mm^2^·s^−1^) represent the thermal conductivity, density, specific heat and thermal diffusivity of the film, respectively. Meanwhile, it has been reported that the thermal conductivities of the composites can be predicted by several physical models such as the Agaris equation [46], effective medium theory (EMT) [47] and Foygel’s theory [48], etc. For the purpose of comparison, in this study, we choose the Agaris equation (i.e., Equation (2)) which is a classic model fitting the composite with low filler loading for estimating the thermal conductivity of particulate composites [1]:log*K* = *V_f_* × C_2_ × logK_f_ + (1 − *V_f_*) × log(C_1_ × K_p_)(2)
where *V_f_* is the volume fraction of filler. K_p_ and K_f_ are the thermal conductivities of polymer substrate and filler, here, they are 0.413 and 350 W·m^−1^K^−1^ [49], respectively. C_1_ is a measure of the factor of filler on the secondary structure of polymer substrate, such as crystallinity and crystal size, which is related to the thermal conductivity of polymer substrate. Referring to a previous study [46], C_1_ can be considered to be 1 in this work; C_2_ measures the degree of difficulty to form conductive pathways and generally should be 0~1. The higher the value of C_2_, the easier to form the conductive pathways. The thermal conductivities of experimental and predicted values of Agari’s model are shown in Figure 6b by setting C_2_ as 1. It can be seen that the thermal conductivity of the composite film prepared by both methods increased with the increase of h-BN. When h-BN content was 23.08 wt%, the thermal conductivities of the blended composite film and aerogel perfusion film were 0.678 W·m^−1^K^−1^ and 1.488 W·m^−1^K^−1^ at 25 °C, respectively, 64.2% and 260.3% higher than that of pure CNF film. Obviously, the predicted values are slightly lower than the experimental values of h-BN/CNF aerogel perfusion-prepared film. Accordingly, the calculated C_2_ should be much higher than 1 if we use Agaris model to fit the measured thermal conductivities of h-BN/CNF aerogel perfusion composite film. These results demonstrated that the perfusion-prepared aerogel h-BN/CNF composite film facilitated the formation of thermally conductive pathways with the low loading of h-BN. To further elucidate the extent of improvement, a parameter *η* was introduced, Equation (3) which is defined as:*η* = (*K_h_* − K_c_) × 100/K_c_(3)
where *K_h_* (W·m^−1^K^−1^) and K_C_ (W·m^−1^K^−1^) represent the thermal conductivity of the composite film and pure CNF film, respectively. As shown in Figure 6c, it can be found that the increment of thermal conductivity improved with the increasing of h-BN. Moreover, the increased range from aerogel perfusion method was much greater than that from blended method. After calendering, the density difference between the two types of film was not significant, but the thermal conductivity of the h-BN/CNF aerogel perfusion composite film was much higher than h-BN/CNF blended composite film. This further demonstrates that perfusion h-BN into CNF aerogel can create more thermally conductive pathways, which is not only conducive to phonon propagation, but also effectively improve the overall thermal conductivity of the composites. It is always desirable to have the higher thermal conductivity with the loading of h-BN as low as possible. As a result, the loading of h-BN above 23.08% was not further conducted.

For electrical insulation applications, a high electrical resistivity of the composite material is important. Figure 6d presents the volume electrical resistivity (*Φ*) of the h-BN/CNF composite films. One can see that the addition of h-BN was helpful to increase the electrical resistivity of the films, and the volume resistivity of the blended composite film was slightly higher than that of the aerogel perfusion composite film at the same h-BN loading. It is possible that the h-BN particles in the h-BN/CNF composite film prepared by the blending method were more evenly dispersed and contributed more towards the insulation performance. As can be seen from Figure 6d, the volume resistivities of pure CNF film, blended h-BN/CNF composite film and aerogel perfusion h-BN/CNF composite film with h-BN loading of 23.08 wt % were 2.68 × 10^14^ Ω·cm, 3.98 × 10^14^ Ω·cm, and 3.83 × 10^14^ Ω·cm, respectively. The h-BN/CNF composite films can fully meet the insulation requirements (Φ > 10^9^ Ω·cm) [50] for electrical insulation applications.

### 3.4. Thermal Management Capability of the h-BN/CNF Aerogel Perfusion Composite Films

In order to demonstrate the thermal management application of the h-BN/CNF aerogel perfusion-prepared composite film, the variations of the surface temperature of the composites with time during heating and cooling were recorded using an infrared thermal imager. To do this, the samples of pure CNF film, h-BN/CNF aerogel perfusion composite film loaded with 23.08 wt % h-BN were vertically placed on the same stage. The thickness of both specimens is the same, i.e., 0.1 mm. The temperature distribution images with time and optical photographs are shown in Figure 7a, in which all the samples were placed in an 80 °C oven for 6 h to ensure uniform sample temperature and then transferred to a thermal insulating wooden stage in a room temperature environment. During the heat dissipation process, the h-BN/CNF aerogel perfusion composite film exhibited much faster decrease with time in comparison with the pure CNF film. 2 s after the sample was placed in the wooden stage, the center temperatures of pure CNF film and h-BN/CNF aerogel perfusion composite film were decreased from 80 °C to 67.9 °C and 56.7 °C, respectively. After 5 s, the films center temperatures reached 41.8 °C and 37.2 °C, and further dropped to 20.9 °C and 19.0 °C after 30 s, respectively. The results indicated that h-BN/CNF aerogel perfusion composite film has good transient heat dissipation performance at high temperature.

The practical thermal management performance of pure CNF film and h-BN/CNF aerogel perfusion composite film with 23.08 wt % h-BN loading was demonstrated in the experiment. A 1 w LED chip with a diameter of 30 mm was applied in the test processing, and every film was cut into a disk-shape with a diameter of 30 mm. In the working process, the pure CNF film and h-BN/CNF aerogel perfusion composite film were put on a flat wooden stage, respectively, and the LED chip was fixed on the top of the film tightly, to ensure the full contact between the film and the LED chip. The surface temperature of the LED chip was recorded in a timely way through an infrared thermal imager. As shown in Figure 7b, the LED chip surface center temperature was 61.4 °C after continue to work for 1 h, while the center temperatures of the LED chip with the pure CNF film and h-BN/CNF aerogel perfusion composite film were 59.1 and 52.1 °C, respectively. h-BN/CNF aerogel perfusion composite film displayed better heat dissipation efficiency in practical applications. The excellent thermal management capability can timely dissipate of the heat within the LED, eventually resulting in an extended lifetime and improved the thermal management efficiency of the electronic devices.

## 4. Conclusions

h-BN/CNF composite film with improved thermal conductivity and electrical insulation properties was successfully fabricated by the perfusion of micro-nano h-BN into CNF aerogel. At 23.08 wt % of h-BN content, the thermal conductivities of the h-BN/CNF aerogel perfusion composite film was 1.488 W·m^−1^K^−1^, a 260.3% increase compared with pure CNF film. The volume resistivities of pure CNF film and h-BN/CNF aerogel perfusion composite film with 23.08 wt % h-BN loading were 2.68 × 10^14^ Ω·cm and 3.83 × 10^14^ Ω·cm, respectively. The thermal management capacity in practical applications also is investigated. After continuing to work for 1 h, the center temperature of the LED chip with the h-BN/CNF aerogel perfusion composite film was 7 °C lower than that of the LED chip with the pure CNF film, indicating that the heat dissipation efficiency of h-BN/CNF aerogel perfusion composite film was better than pure CNF film. Therefore, the as-fabricated h-BN/CNF aerogel perfusion composite film is promising to replace the traditional synthetic polymer film materials as a green, eco-friendly and flexible thermally conductive and electrical insulation film, leading to a broad potential application in the field of electronics.

## Figures and Tables

**Figure 1 nanomaterials-09-01051-f001:**
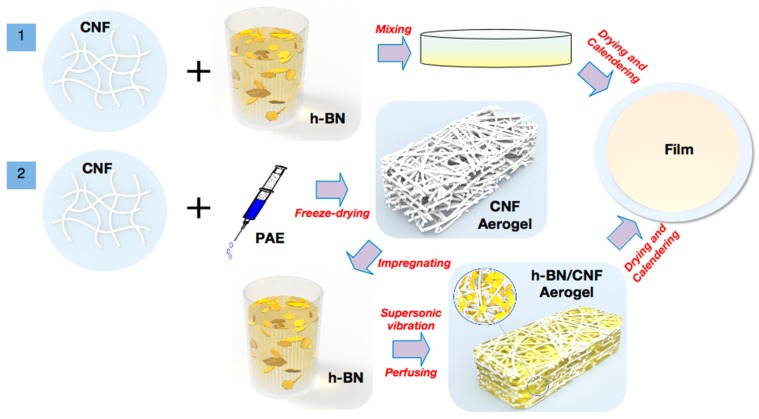
Schematic of preparation of hexagonal boron nitride/cellulose-nano fiber (h-BN/CNF) composite film.

**Figure 2 nanomaterials-09-01051-f002:**
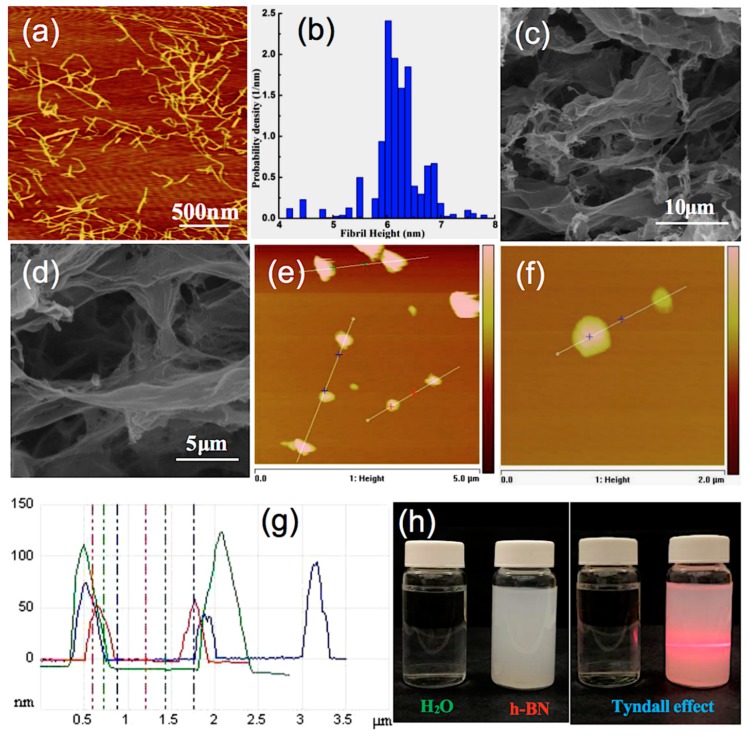
(**a**) Atomic force microscope (AFM) image of CNF. (**b**) The corresponding height distribution measured from AFM topographic data of CNF. (**c**,**d**) Scanning electron microscope (SEM) images of CNF aerogel under different magnifications. (**e**) and (**f**) AFM images of h-BN under different magnifications. (**g**) The corresponding height distribution measured from AFM topographic data from Figure e. (**h**) Tyndall phenomenon of h-BN suspension.

**Figure 3 nanomaterials-09-01051-f003:**
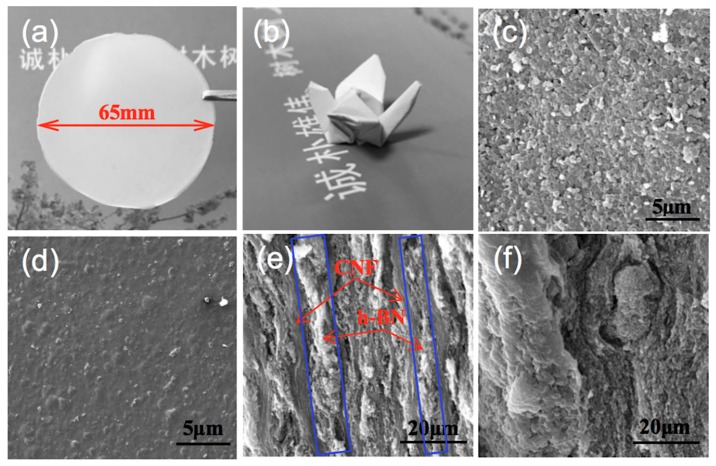
(**a**) Optical photograph of h-BN/CNF aerogel perfusion composite film with 23.08 wt % h-BN loading, red arrow indicated the diameter of the film was 65 mm. (**b**) Optical photograph of a “paper crane” folded from a h-BN/CNF aerogel perfusion composite film. (**c**,**d**) Surface SEM images of h-BN/CNF aerogel perfusion composite film and h-BN/CNF blended composite film with 23.08 wt % h-BN loading, respectively. (**e**,**f**) Cross-sectional SEM images of h-BN/CNF aerogel perfusion composite film and h-BN/CNF blended composite film with 23.08 wt % h-BN loading, respectively, red arrows were channels of CNFs and h-BN, blue rectangles were thermally conductive pathways of h-BN.

**Figure 4 nanomaterials-09-01051-f004:**
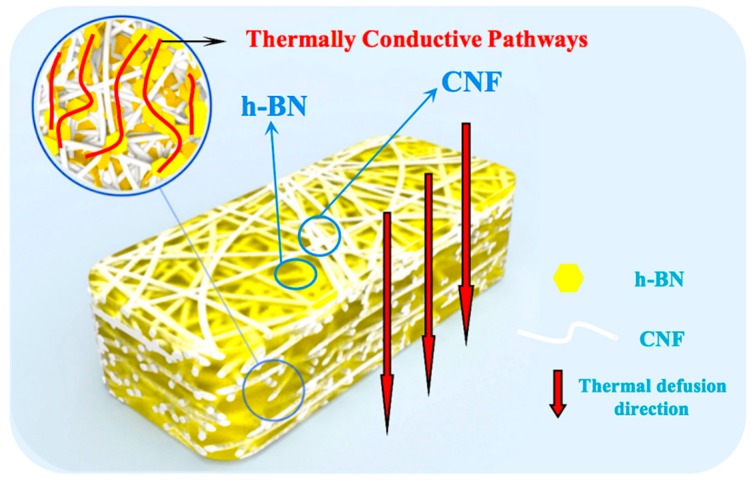
The proposed thermal defusion mechanism of h-BN/CNF aerogel perfusion film.

**Figure 5 nanomaterials-09-01051-f005:**
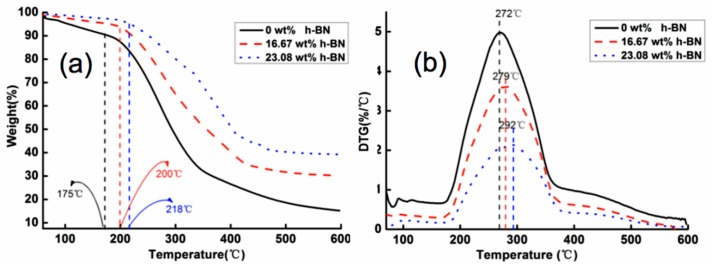
(**a**) Thermogravimetric (TG) and (**b**) Derivative thermogravimetry (DTG) curves of h-BN/CNF aerogel perfusion composite films.

**Figure 6 nanomaterials-09-01051-f006:**
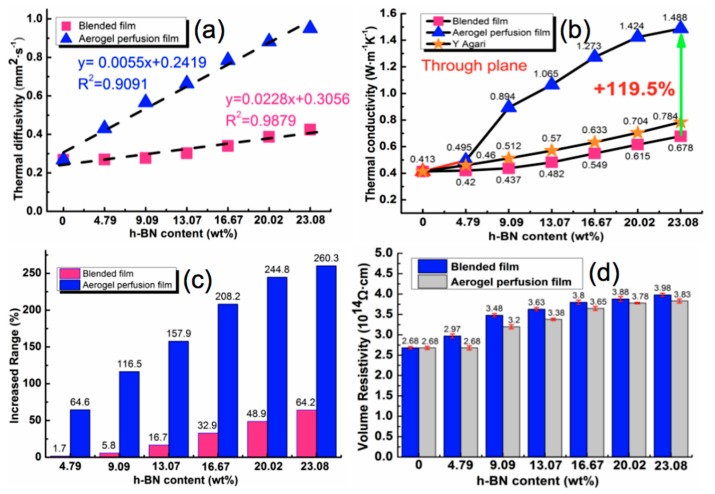
The effects of h-BN loading on (**a**) Thermal diffusivity, (**b**) Thermal conductivity, (**c**) Increased range of thermal conductivity, (**d**) The effects of h-BN loading on volume resistivity.

**Figure 7 nanomaterials-09-01051-f007:**
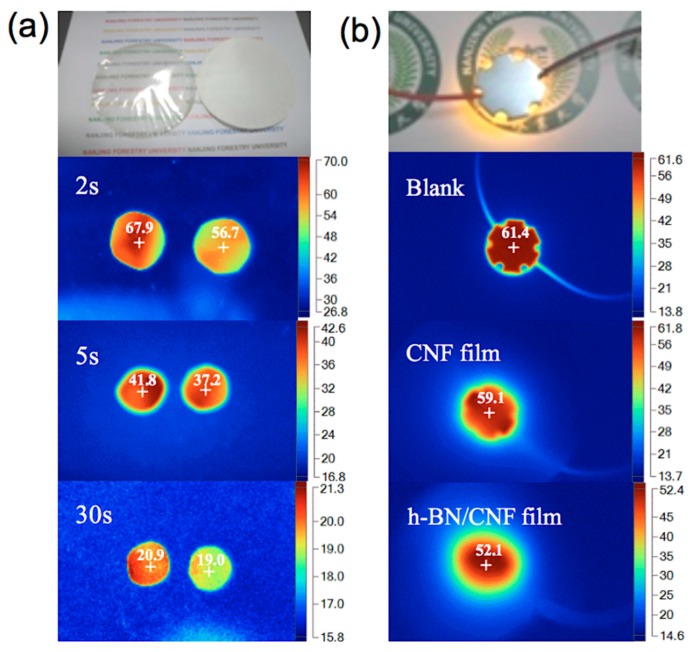
(**a**) Thermographs of pure CNF film and h-BN/CNF aerogel perfusion composite film with 23.08 wt % h-BN loading in cooling process. (**b**) Thermographs of pure CNF film and h-BN/CNF aerogel perfusion composite film with 23.08 wt % h-BN loading on the surface of a light-emitting diode (LED) luminescence chip.

**Table 1 nanomaterials-09-01051-t001:** Ratio of h-BN of the composite films (the CNF dry weight of all samples was 0.167 g).

Sample	h-BN (g)	h-BN Loading (wt %)	h-BN Loading (vol %)
1	0	0	0
2	0.0084	4.79	1.59
3	0.0167	9.09	3.18
4	0.0251	13.07	4.78
5	0.0334	16.67	6.33
6	0.0418	20.02	7.92
7	0.0501	23.08	9.51

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
