# Peer review of "Aerogel Perfusion-Prepared h-BN/CNF Composite Film with Multiple Thermally Conductive Pathways and High Thermal Conductivity"

_nanomaterials, 2019, doi:10.3390/nano9071051_

Reviewer 1 Report

The manuscript (Manuscript Number nanomaterials-549529) illustrates method for the synthesis of perfusing hexagonal boron nitride (h-BN) into cellulose-nano fiber (CNF) (100~2000 nm in length,<100 nm in diameter) aerogel originating from plant fibers to create multiple thermally conductive pathways of the composite film. As authors stated, this procedure is important, because it not only overcomes the drawback of high resistance between ordinary plant fibers and thermal conducting particles, but also makes the film thinner, leading to a flexible, foldable, high temperature and aging resistant insulating material. Therefore, the resulting h-BN/CNF film is very promising to replace the traditional synthetic polymer materials for a broad spectrum of applications, including the field of electronics.

On the whole, the manuscript is fairly well-written and logically arranged. The overall originality of the review concept used here is medium. Nevertheless, I would recommend publication of this review article in Nanomaterials on the condition a minor revision of the manuscript will be carried out and the following points will be taken into consideration.

Detailed comments:

1.      The abstract needs to be well written with future prospects of the work and describe in short the concept of the fabrication of h-BN thermally conductive pathways, especially in the context of 3D cellulose aerogel skeleton.

2.      There are no recent and equally valuable references related to fabrication of h-BN in the Introduction.  Furthermore, the introduction should be worked out - so as to show the full state of knowledge on this topic. Extension to look at these issues and also provide other electrodeposition techniques should also be provided.

3.      More detailed results discussion should be provided. The chapter appears to be a collection of data from research papers, however, authors self-opinion is of importance while drafting a chapter of this type.

4.      The conclusion reflects an overall summary of the field with further extension and includes future prospective - I would suggest clarifying this section.

After completing the above-mentioned corrections this work will be more readable. Therefore, it will be useful for the readers of Nanomaterials.

Author Response

Point 1: The abstract needs to be well written with future prospects of the work and describe in short the concept of the fabrication of h-BN thermally conductive pathways, especially in the context of 3D cellulose aerogel skeleton.

Response 1: Thanks for the suggestion. The revision has been made in the Abstract”. We add h-BN based materials application prospects and point out the existing problems in the preparation of thermally conductive materials. We also describe the concept of the fabrication of h-BN thermally conductive pathways shortly. Please see the manuscript.

Point 2: There are no recent and equally valuable references related to fabrication of h-BN in the Introduction.  Furthermore, the introduction should be worked out - so as to show the full state of knowledge on this topic. Extension to look at these issues and also provide other electrodeposition techniques should also be provided.

Response 2: The advice of adding recent and valuable references related to fabrication of h-BN in the Introduction is of great help to the improvement of the article. We have added the part of the fabrication of h-BN and cited the recent relevant papers in the revised manuscript. The “Introduction” has been modified carefully and highlighted in yellow. Please see the revised manuscript.

Point 3: More detailed results discussion should be provided. The chapter appears to be a collection of data from research papers, however, authors self-opinion is of importance while drafting a chapter of this type.

Response 3: Thanks for the suggestion. We have added more detailed results discussion in our manuscript, which highlighted in yellow. Please see the revised manuscript.

Point 4: The conclusion reflects an overall summary of the field with further extension and includes future prospective - I would suggest clarifying this section.

Response 4: We have revised this section, please see the manuscript.

Finally, the comments and suggestions from the reviewers are highly appreciated.

Reviewer 2 Report

Dear authors!

Your study on the thermal and electrical conductivity of CNF - h-BN composites is quite interesting from a scientific point of view.

Anyhow there are several issues that would need be improved considerably:

1) The article does not hold much novel information compared to your previous study "Polymers 2019, 11, 660 by Xiu Wang et al, doi:10.3390/polym11040660" of a similar or even the same material with the same methods. If there would be novelty it is not clearly visible or described. A discussion with respect to the first published work is missing.

2) The introduction mainly and the article partyl hold citations which are not appropriate. This means that the argument or information given with a citation is not found or not given in the same meaning or even value in the citation. Also citations are not fully given thus often a certain infromation like Issue is missing and several issues have to be checked befor fining the proper publication.

3) Several figures of the supplemental information is taken from your previous work and not cited properly, i.e. Figure S3 b, c, Figure S4 and Figure S5.

4) Figure S4 and Figure S5 are indicated with 0, 16.67 and 23.08 w% although the spectra clearly are the same as in your Polymers article but there they are given with up to 40 w%. This is not only not citing but a clear inconsistency that i can not explain.

There are sevearl issues in the article besides this:

- All °C and other type setting of special letters is spoiled

- Line 17-18 reformulate

- line 25 "make a device" - change formulation

- line 27 "burning and explosion" citations 1 and 4 hold totally different content, they do not speak of this given risks

- Line 32 citation 15 speks of PA only...

line 42 citation 23 deals with thinner films

line 44 citation 24 claims 20-50 µm but anyhow without reference

line 90 reformulate sentence

line 96 what is a "wet strength function"

line 117: It is stated that moisture was removed at 25°C for 5h, though aerogels are usually not as easily dryied since they absorb rather humidity.

line 138 the formula is not displayed correctly

line 151-152 the Tyndall effect as criterion for siez of<100µm is unclear since only red laser light is used and the more the citation 29 is only talking about this "Rayley scattering" in general terms. Basic optical literature might be more appropriate. Anyhow the relation to the size of particles is not clear. Also line 174 with the reference to Fig2h!

Figure 2b is known from "Polymers" publication

line 166 Phonon carrier to reduce phonon scattering is not clear

Figure 4 - line 220: Figure 3e indicates orientation of the CNF and parallel orientation of the h-BN but Figure 4 states h-BN pathways orthogonal to it, how does this fit?

line 230 reformulate

line 239 if the h-BN only decomposes at 800°C it is thus assumed to be still intact at the TG?

line 243 what is "thermal motion" of CNF and how does it affect the decompostion of it?

Figure 5 the onset of the TG durve of 0w% seems rather near 200 and thus also closer to the literature value expected which would only indicate that 16.67w% does not have a considerable effect.

line 296 and text: obviously a wrong text or wrong equation is given, what is Kp and C1?

line 309 if 0<C2<1 how could C2 possibly be much higher than 1 - this would only indicate that the model is not appropriate.

line 313 something is missing after "a parameter"

line 334: might the resistivity be dependent on the thickness of the film? i.e. are there surface effects in the composite which might influence applicability for thin layers? How thick a layer is usually used?

line 339-341 and citation 42: this is not state in the literature. it rather speaks of "All the composites containing the two kinds of thermally conductive fillers present a characteristic of the insulator (Φ>   109Ω ·cm)" - where exactly this limit comes from is also not explained in this citation.

line 353 if "pure CNF film and h-BN/CNF" both show rapid decreas in center temperature, where is the benefit of loading with h-BN?

line 361-364 the explanation is not clear. If the paper is on the LED and condcuting heat well, it should display hotter. But then air stream and surface roughness will play a crucial role! If the paper is below the LED is might get rid of some heat towards the substrate below. The stacking what is on top of what and the statement istelf are not clear.

Author Response

Point 1: The article does not hold much novel information compared to your previous study "Polymers 2019, 11, 660 by Xiu Wang et al, doi:10.3390/polym11040660" of a similar or even the same material with the same methods. If there would be novelty it is not clearly visible or described. A discussion with respect to the first published work is missing. 

Response 1: The approach used in our work for preparing the composites is rather different from the one reported in “Polymers 2019, 11, 660, Xiu Wang et al”. 

In fact, we used CNFs to prepare aerogels firstly, and the wet strength agent, poly(aminoamide)-epichlorohydrin (PAE) resins, were added to enhance the wet strength of aerogel. Afterward, CNF aerogels were impregnated into a certain amount of h-BN suspension with ultrasonic-assisted mixing; followed by drying and calendering, the h-BN/CNF aerogel perfusion composite films were then fabricated. The method reported in “Polymers” referred to mixing CNF and BNNS (1:3) first to fabricate BNNS/CNF aerogel, and then filled with CNF to prepare the composites. During our process, PAE was used, and the components of the aerogel are much difference between the two papers, which also clearly differs our work from that reported previously. 

The methodology developed for the preparation of h-BN/CNF composite film represents one of the key novelties of our work. We have compared to our previous study. Please see the manuscript.

Point 2: The introduction mainly and the article partly hold citations which are not appropriate. This means that the argument or information given with a citation is not found or not given in the same meaning or even value in the citation. Also citations are not fully given thus often a certain information like Issue is missing and several issues have to be checked before fining the proper publication.

Response 2: We have checked the references carefully and made appropriate modifications. Please see the manuscript.

Point 3: Several figures of the supplemental information is taken from your previous work and not cited properly, i.e. Figure S3 b, c, Figure S4 and Figure S5. 

Response 3: We want to compare the size difference between micro-nano h-BN and BNNSs, we have revised the manuscript and cited properly. For Figure 3b and c, the exfoliation method and equipment in both two articles are same, i.e., the h-BN suspension was sonicated in a sonication bath (X0-650, Xianou Instruments Ltd., China) for 8 h with a frequency of 200 kHz and a 520 W output power. The difference between them was the centrifugal speed. Please see the revised manuscript.

Point 4: Figure S4 and Figure S5 are indicated with 0, 16.67 and 23.08 w% although the spectra clearly are the same as in your Polymers article but there they are given with up to 40 w%. This is not only not citing but a clear inconsistency that i can not explain.

Response 4: The CNF and h-BN used in two papers are the same, so the functional groups and crystalline in both two raw materials should be in common. The difference between them is the loading of CNF and BN. The table of BNNSs and CNF amounts of the composite films published in “Polymers” is shown below (please see the word file of response). The weight of BNNSs is much similar. We can see that the weight of 40 wt% of BNNS is 0.050 g, same as the 23.08 wt% in this work. So the Figure S5 looks much similar to Polymers’ article. What’s more, the ordinate range in the two articles is different, which further leads to the similarity between the two figures. As for the FT-IR spectrum in Figure S4, has the same problem. After careful consideration, for this paper, we think that the thermal conductivity and physical model of the composite film are much important than the chemical characteristics, so we're going to delete this part from the manuscript. Please see the manuscript.

Point: There are several issues in the article besides this:

- All oC and other type setting of special letters is spoiled

Response: We have fixed all the type setting of special letters.

- Line 17-18 reformulate

Response: Have reformulated in the manuscript and highlighted.

- line 25 "make a device" - change formulation

Response: Have changed.

- line 27 "burning and explosion" citations 1 and 4 hold totally different content, they do not speak of this given risks

Response: Thanks for your meaningful suggestions. We have revised the citations in our manuscript. The references titled  “Thermally Conductive and Electrical Insulation BNNS/CNF Aerogel Nano-Paper” and “A combination of boron nitride nanotubes and cellulose nanofibers for the preparation of a nanocomposite with high thermal conductivity”, i.e.,  the references 3 and 4, come up with the idea of the explosion in the abstract and introduction clearly. So we retain references 3 and 4 to support this idea. Now is referenced 6 and 7. Please see the manuscript.

- Line 32 citation 15 speaks of PA only...

Response: We have added more references related to the embrittlement and revised in the manuscript.

- line 42 citation 23 deals with thinner films

Response: As shown in below figure from citation 23 (please see the word file of response), we can see that the thickness of PI films applied in different fields related to electronics are basically less than 125 μm, and we summarize it as “the thicknesses of electronic insulating film materials such as polyimide films are often below 125 μm, it is appropriate in this case.

- line 44 citation 24 claims 20-50 μm but anyhow without reference

Response: We have cited a more convincing reference in place of the original reference, i.e., “Ververis, C.; Georghiou, K.; Christodoulakis, N.; Santas, P.; Santas, R. Fiber dimensions, lignin and cellulose content of various plant materials and their suitability for paper production. Ind. Crops Prod. 2004, 19, 245-254”. Please see the revised manuscript.

- line 90 reformulate sentence

Response: Have reformulated in the manuscript.

- line 96 what is a "wet strength function"

Response: Generally, the physical strength of hygroscopic materials will be reduced when they absorb a certain amount of water. The strength of the fiber measured in the wetting state (when it is soaked in water to achieve complete equilibrium) is wet strength. After adding a wet strength agent, poly(aminoamide)-epichlorohydrin (PAE) resins, the wet strength of CNF aerogel will be improved. We have revised the sentence “CNF aerogels with wet strength function were obtained after heated at 105 oC for 30 min to induce cross-linking.” to “CNF aerogels with enhanced wet strength were obtained after heated at 105 oC for 30 min to induce cross-linking.” Please see the revised manuscript.

- line 117: It is stated that moisture was removed at 25oC for 5h, though aerogels are usually not as easily dried since they absorb rather humidity.

Response: A lot of water goes into the CNF aerogel after perfusion h-BN suspension. There are two main steps of dehydration to obtained the dry film. Firstly, transferred composites into a blast oven at 25 oC for 5 h to remove a part of the water. Secondly, the aerogel composites after preliminary dehydration were placed between two PVDF membranes to compact under 0.5 MPa pressure, to achieve the purpose of “squeeze de-watering”. So the purpose of the first step is that dehydrated a part of the water, and the dry composite film will be fabricated after the second step. Please see the revised manuscript.

-line 138 the formula is not displayed correctly

Response: The formula has been completed, please see the manuscript.

- line 151-152 the Tyndall effect as criterion for size of<100 μm is unclear since only red laser light is used and the more the citation 29 is only talking about this "Rayley scattering" in general terms. Basic optical literature might be more appropriate. Anyhow the relation to the size of particles is not clear. Also line 174 with the reference to Fig2h!

Response: We have approved that the length and diameter of the CNFs are 300-1000nm and 6-7 nm by AFM, respectively. We didn’t use the Tyndall effect to explain the size distribution of CNFs, but because the size of CNFs is in this range, and lead to the Tyndall effect. We have cited a more convincing reference in place of the original reference and added a reference related to Figure 2h. Please see the revised manuscript.

- Figure 2b is known from "Polymers" publication

Response: It is rather different from the one reported in “Polymers”, as shown in the below figure (please see the word file of response).

- line 166 Phonon carrier to reduce phonon scattering is not clear

Response: Thermal conduction of non-metallic fillers (such as BN, AlN, SiC, etc.) mainly depends on the phonon vibration to transfer the energy. It is an effective way to increase the contact area and decrease the inter-layered gap to reduce the phonon scattering between h-BN. In our work, micro-nano h-BN was obtained by exfoliation method. The contact area between h-BN was improved and leading to a relatively gentle phonon scattering at the interfaces of h-BN, it is beneficial to the thermal conductive efficiency. Please see the revised manuscript.

- Figure 4 - line 220: Figure 3e indicates orientation of the CNF and parallel orientation of the h-BN but Figure 4 states h-BN pathways orthogonal to it, how does this fit?

Response:  It can be known that the CNF aerogel has a typical 3D porous structure with a pore diameter of 5~20 μm. After impregnation by h-BN suspension, the pores of CNF aerogel were fully filled with h-BN. After drying and calendering, the layered structure of h-BN and CNF could also be observed in h-BN/CNF composite film, as shown in Figure 3e. Figure 4 illustrated the proposed morphology of h-BN/CNF aerogel before squeezed and calendered, not h-BN/CNF film, it is the precursor of the composites shown in Figure 3e. The porous CNF aerogels were filled with h-BN suspension, the white filaments represent CNFs, h-BN is shown in a yellow hexagon, h-BN contacts with each other to form thermally conductive pathways, which red lines represent. After calendering, the “white foam” shape h-BN/CNF aerogel will be squeezed and formed the layer structure.

- line 230 reformulate

Response: Have reformulated, please see the revised manuscript.

line 239 if the h-BN only decomposes at 800oC it is thus assumed to be still intact at the TG?

Response: Yes, h-BN is very stable and it doesn't break down even at 800 oC. The TG curve of BN is shown in the following figure (please see the word file of response).

- line 243 what is "thermal motion" of CNF and how does it affect the decompostion of it?

Response: “Thermal motion” means “Thermal decomposition” in this case, it is the process by which compounds are broken down by heating. In our work, h-BN/CNF 3D aerogel skeleton was constructed, a portion of CNF was wrapped between layers h-BN. And from Figure 3e we know that the composite film has an h-BN/CNF layer-by-layer structure, h-BN closely connected with each other and provided a tortuous path for the diffusion of gas molecules and significantly reduced the permeation rate of gas. Herein, the oriented BNNS platelets acted as a physical barrier and delayed the escape of degradation products, which finally promoted the thermal decomposition temperature to move towards high temperatureThus the thermal stability of the composites was improved. We have revised in the manuscript and highlighted.

- Figure 5 the onset of the TG curve of 0w% seems rather near 200 and thus also closer to the literature value expected which would only indicate that 16.67w% does not have a considerable effect.

Response: Analyzing from another angle. We can see from Figure 5a, at 200 oC, the weight of 0 wt% h-BN film is 88% while 16.67 wt% h-BN film is 95% approximately. And from Figure 5b, the DTG (%/oC) of 0 wt% h-BN film is 5 while 16.67 wt% h-BN film is 3.5. This fully indicates that adding 16.67 wt% h-BN can increase the stability of the composite film.

- line 296 and text: obviously a wrong text or wrong equation is given, what is Kp and C1?

Response: We have revised the equation in the manuscript.

- line 309 if 0<C2<1 how could C2 possibly be much higher than 1 - this would only indicate that the model is not appropriate.

Response: A calculated C2 should be 1.80 if we use Agari’s model to fit the measured thermal conductivity of composite film with 1.59 vol% h-BN. Although the maximum difference of thermal conductivity is about 0.7, the distinction of C2 is not significant after calculation according to the equation (2). Moreover, this model has been verified in the paper which has already published in Advanced Functional Materials named “Cellulose nanofiber supported 3D interconnected BN nanosheets for epoxy nanocomposites with ultrahigh thermal management capability”. The method to prepare the composites is much similar with us. So we think this model is appropriate in our work. What’s more, C2 measures the degree of difficulty to form conductive pathways, the higher the value of C2, the easier to form the conductive pathways, this also indicates that it is easier to form thermal conductive pathways in our study, which verifies the concept in our study.

- line 313 something is missing after "a parameter"

Response: We have added “h” after “a parameter”. 

- line 334: might the resistivity be dependent on the thickness of the film? i.e. are there surface effects in the composite which might influence applicability for thin layers? How thick a layer is usually used?

Response: The thickness of both specimens is 0.1 mm approximately. The insulation performance of materials is affected by many aspects, such as humidity, moisture content, temperature, etc. In our work, the films were obtained by calendering, so the surface morphology, such as roughness, we considered it to be the same in all samples. And in our work, the volume resistance was tested at 25 oC and 50% humidity, we recognized that the external effects have the same effect on the film. We can calculate the volume of electrical resistivity (Φ) according to the equation as follows: Φ=R´Ae/t. while Φ is the volume resistivity, R is the resistance, Ae is the area of the electrode, t is the thickness of the sample.

- line 339-341 and citation 42: this is not state in the literature. it rather speaks of "All the composites containing the two kinds of thermally conductive fillers present a characteristic of the insulator (Φ>109 Ω·cm)" - where exactly this limit comes from is also not explained in this citation.

Response: We have added a more convincing reference in the manuscript.

- line 353 if "pure CNF film and h-BN/CNF" both show rapid decrease in center temperature, where is the benefit of loading with h-BN?

Response: As we can see from Figure 7a, the temperature of the h-BN/CNF film exhibited much faster decrease with time in comparison with the pure CNF film. 2 seconds after, the center temperatures of pure CNF film and h-BN/CNF aerogel perfusion composite film were 67.9 oC and 56.7 oC, respectively. Due to the large difference between the h-BN/CNF film, pure CNF film, and the indoor temperature at the beginningso the temperature drops fast in both kinds of films. But, 2s after, the center temperature of h-BN/CNF film is 56.7 oC and the pure CNF film is 67.9 oC, the difference was as much as ~11 degrees, so adding h-BN can improve the heat dissipation property of the film obviously. We deleted the “rapidly” for avoiding ambiguity. Please see the manuscript.

- line 361-364 the explanation is not clear. If the paper is on the LED and conducting heat well, it should display hotter. But then air stream and surface roughness will play a crucial role! If the paper is below the LED is might get rid of some heat towards the substrate below. The stacking what is on top of what and the statement itself are not clear.

Response: The function of the h-BN/CNF thermal conductive film is to transfer the heat generated by the electronic device timely rather than store it. Obviously, h-BN/CNF composite film can reduce the operating temperature of the electronic devices and have better heat dissipation performance than pure CNF film. Our focus is to study the heat dissipation property of two kinds of composite films, and the testing condition is the same, so we ignored the effect of the air stream. The preparation of the composite film has gone through the process of calendering, so we also ignore the effect of roughness on the composite film. And we know that the main effect of heat dissipation efficiency is the loading of thermal conductivity filler: h-BN. Since the test environment is the same, the influence of various factors on the two kinds of film exists simultaneously, which is fair to the two kinds of film. We have reformulated the sentence and revised the manuscript. Please the highlighted part in the manuscript.

Finally, the comments and suggestions from the reviewers are highly appreciated.

Round  2

Reviewer 1 Report

I would like to support this revised paper (Manuscript Number nanomaterials-549529) for the publication in Nanomaterials. All suggested changes were made (or discussed/clarified) by the authors. The results are informative, and discussion is clear. Summarizing, I think that this paper can be published as is.

Author Response

We are very grateful to the comments, which are not only very helpful in improving the quality of our manuscript, but also beneficial to our work significantly.

Reviewer 2 Report

Dear Authors,

thank you for undergoing such an effort to improve the manuscript. Many passages clearly are enhanced. Three things still should be considered.

1) The two histrograms very much resemble each other, despite the fact the the scale has been extended - the height of all but the two highest bars beeing neraly exactly the same, indeed only these two histrogram bars are different is statistically very unlikey. Is the reproducibility this high in your sample, measurement and stattistics as to produce the very like propability pattern?  I highly recommend if the same batch has been used to cite not only the Figure S3 b and c image but clearly also the Polymers publication for the histrogram or to comment on this issue. Or was a different subset evaluated changing only some bars slightly? It might as well be a question of the region or the total area or number of CNF considered. It seems strange anyhow and should be clearified.

2) In the presented version of the supplemental information Figure S3 b and c i can still not see the citation of Figure 1 of the Polymers publication. Was it forgotten or not displayed correctly?

3) The heat dissipating capacity of your film is now much nicer descirbed with the LED setting. Still I was puzzled how does it work. On the table (or substrate) there is either directly the LED or the film and the LED on top. Thus if the size of the underlaying film is exactly that of the LED (is this the case?) the constant flow of heat downwards away from the LED and thus cooling the device is mainly limited by the table (or subrate). In this case the LED directly on the table might as well suffer from a tiny air gap giving rise to an additional insulation and the application of the film would breach this gap. On the other hand a much bigger film below the LED might very well dissipate the heat along the film to a much bigger surface from where cooling off. I want to point out that this test misses still more description in order to be fully understandable. A more accurate description of this experiment would be appropriate in order to fully be able to grasp the idea of this qualitative showing.

Author Response

Point 1: The two histrograms very much resemble each other, despite the fact the the scale has been extended - the height of all but the two highest bars beeing neraly exactly the same, indeed only these two histrogram bars are different is statistically very unlikey. Is the reproducibility this high in your sample, measurement and stattistics as to produce the very like propability pattern?  I highly recommend if the same batch has been used to cite not only the Figure S3 b and c image but clearly also the Polymers publication for the histrogram or to comment on this issue. Or was a different subset evaluated changing only some bars slightly? It might as well be a question of the region or the total area or number of CNF considered. It seems strange anyhow and should be clearified.

Response 1: In both two paper, we use the same CNF that prepared by the same method and same bath, so the diameter of the CNF was very similar. To ensure the obvious distinction, we choose another AFM picture of CNFs which taken at the same time, and the corresponding height distribution was recalculated. Please see the revised manuscript.

Point 2: In the presented version of the supplemental information Figure S3 b and c i can still not see the citation of Figure 1 of the Polymers publication. Was it forgotten or not displayed correctly?

Response 2: Thanks for the suggestion. We only added the citation in our manuscript before. Now we have cited the reference in the supplemental information. Please see the revised supplemental information.

Point 3: The heat dissipating capacity of your film is now much nicer described with the LED setting. Still I was puzzled how does it work. On the table (or substrate) there is either directly the LED or the film and the LED on top. Thus if the size of the underlaying film is exactly that of the LED (is this the case?) the constant flow of heat downwards away from the LED and thus cooling the device is mainly limited by the table (or subrate). In this case the LED directly on the table might as well suffer from a tiny air gap giving rise to an additional insulation and the application of the film would breach this gap. On the other hand a much bigger film below the LED might very well dissipate the heat along the film to a much bigger surface from where cooling off. I want to point out that this test misses still more description in order to be fully understandable. A more accurate description of this experiment would be appropriate in order to fully be able to grasp the idea of this qualitative showing.

Response 3: Thanks for your meaningful suggestion. We have revised in the manuscript. In our study, a 1 w LED chip with a diameter of 30 mm was applied, and every film was cut into a disk-shape with a diameter of 30 mm, which was same with the size of LED chip. In the working process, the film was put on a flat wooden stage firstly, and the LED chip was fixed on the top of pure CNF film and h-BN/CNF aerogel perfusion composite film tightly, to ensure the fully surface contact between the film and the LED chip. The order of the unit is: bottom (wooden stage), medium (film), top (LED chip). The surface temperature of the LED chip were timely recorded through an infrared thermal imager. As you said, in our testing process, the air flow will affect the experiment possibly, but our whole process is completed under the same test environment, we try to minimize the error (in many papers, they have ignored the the environmental impacts), this can fully prove that the h-BN/CNF composite film has better heat dissipation efficiency compared with pure CNF film, and that is what we want to prove.

The detailed comments and suggestions from the reviewers are highly appreciated.